# Surface Transformation of Spin-on-Carbon Film via Forming Carbon Iron Complex for Remarkably Enhanced Polishing Rate

**DOI:** 10.3390/nano12060969

**Published:** 2022-03-15

**Authors:** Jun-Myeong Lee, Jong-Chan Lee, Seong-In Kim, Seung-Jae Lee, Jae-Yung Bae, Jin-Hyung Park, Jea-Gun Park

**Affiliations:** 1Department of Nanoscale Semiconductor Engineering, Hanyang University, Seoul 04763, Korea; kjunom200@naver.com (J.-M.L.); student.jongchan@gmail.com (J.-C.L.); rlatjddls25805@naver.com (S.-I.K.); 2Department of Electronic Engineering, Hanyang University, Seoul 04763, Korea; ltwobooml@naver.com; 3Department of Energy Engineering, Hanyang University, Seoul 04763, Korea; jaeyoungb@daum.net; 4UB Materials Inc., Yongin 17162, Korea; parkjinhyung@gmail.com

**Keywords:** chemical–mechanical planarization, spin-on-carbon (SOC), hard-mask, C-C bond breakage, ferric catalyst

## Abstract

To scale down semiconductor devices to a size less than the design rule of 10 nm, lithography using a carbon polymer hard-mask was applied, e.g., spin-on-carbon (SOC) film. Spin coating of the SOC film produces a high surface topography induced by pattern density, requiring chemical–mechanical planarization (CMP) for removing such high surface topography. To achieve a relatively high polishing rate of the SOC film surface, the CMP principally requires a carbon–carbon (C-C) bond breakage on the SOC film surface. A new design of CMP slurry evidently accomplished C-C bond breakage via transformation from a hard surface with strong C-C covalent bonds into a soft surface with a metal carbon complex (i.e., C=Fe=C bonds) during CMP, resulting in a remarkable increase in the rate of the SOC film surface transformation with an increase in ferric catalyst concentration. However, this surface transformation on the SOC film surface resulted in a noticeable increase in the absorption degree (i.e., hydrophilicity) of the SOC film CMP slurry on the polished SOC film surface during CMP. The polishing rate of the SOC film surface decreased notably with increasing ferric catalyst concentration. Therefore, the maximum polishing rate of the SOC film surface (i.e., 272.3 nm/min) could be achieved with a specific ferric catalyst concentration (0.05 wt%), which was around seven times higher than the me-chanical-only CMP.

## 1. Introduction

Recently, in order to achieve faster switching, lower power consumption, and lower bit-cost, nanoscale semiconductor devices have been rapidly scaled down; for example, a design rule less than 14 nm for dynamic random access memory (DRAM), memory cells with more than 128 floors for 3-dimensional (3D) NAND flash memory, and a design rule less than 5 nm for application processors [1,2,3,4,5]. The fabrication of these nanoscale devices involves ArF immersion lithography using a 193 nm ArF excimer laser and extreme ultra-violet lithography (EUVL), and using 13.8 nm laser-produced plasma or synchrotron radiation [6,7]. Using these lithography systems, photoresist patterns with a high aspect ratio of photoresist thickness to photoresist pattern size (i.e., >5:1) are required for assuring a mask role against the dry-etching process [8,9]. However, as photoresist patterns with a high aspect ratio could easily result in a collapse of photoresist patterns, a hard-mask between the photoresist and the substrate film being etched has been introduced for the lithography and etching process, as shown in (i) of Figure 1a. In general, this lithography and etching process using a hard-mask is followed by (i) hard-mask deposition (or spin-coating) on the substrate being etched, (ii) chemical–mechanical planarization (CMP) and cleaning, (iii) SiON layer chemical-vapor deposition (CVD) and photoresist spin coating, (iv) photoresist exposure and development, (v) SiON layer etching, (vi) hard-mask etching, (vii) photoresist strip, (viii) substrate film etching, and SiON strip [10,11]. Thus, the hard-mask supplies the transfer role of the photoresist patterns via anisotropic dry etchings, requiring high chemical, heat, and etching resistance.

Two kinds of hard-masks were applied, i.e., amorphous carbon layer (ACL) and spin-on-carbon (SOC) film [12,13]. ACL was deposited by the CVD process at a high temperature (i.e., >500 °C), presenting a high hardness (i.e., ~6.51 GPa), as shown in Appendix A, demonstrating an excellent anisotropic etching characteristic and a high etching resistance. However, as the ACL process is conducted by CVD at a high temperature, it is confronted by a high generation of organic particles, a high film surface roughness, a difficulty of CMP due to a high hardness, and a high process cost, as shown in Appendix A. As an alternative, SOC film has been applied as a hard-mask, as it can provide a cheap process cost, a low film surface roughness, and a high process flexibility to adjust the film thickness, hydrophilicity, absorbance, and refractive index, because of the spin-coating process of a polyarylene-ether block copolymer solution at room temperature [14,15,16]. Nevertheless, as the hardness of an SOC film (i.e., ~0.63 GPa) is remarkably lower than that of an ACL (i.e., ~6.51 GPa), the SOC film thickness should be relatively thicker than the ACL thickness to assure chemical, heat, and anisotropic etching resistance. Thus, the coating of a thick SOC film on a patterned semiconductor chip fabricated on a 12-inch wafer leads to the dependency of the SOC thickness on the pattern density of the substrate SiO_2_ film, resulting in a severely non-uniform SOC film thickness on the patterned semiconductor chips (patterned substrate SiO_2_ film), as shown in (i) and (ii) of Figure 1b. Therefore, a long-wavelength CMP is required for eliminating the dependency of the SOC thickness on the pattern density of the substrate SiO_2_ film. Note that a long-wavelength CMP should simultaneously conduct a surface SOC film topography planarization in both areas of high and a low pattern density, as shown in (iii) of Figure 1b. 

Generally, an SOC film is composed of a polyarylene-ether block copolymer [17,18,19]. Because of the strong covalent carbon–carbon (C-C) bonds (sp^2^ or sp^3^) in a polyarylene-ether block copolymer, the polishing rate of an SOC film during CMP is relatively low, i.e., ~27.7 nm/min, as shown in Figure 2. Thus, such a low SOC film-polishing rate cannot perform a complete planarization of the surface topography of the SOC film. As a solution, a novel concept to break the strong covalent C-C bonds on the SOC film surface should be introduced, as the event of breaking the strong covalent C-C bonds on the SOC film surface during CMP can remarkably enhance the polishing rate of the SOC film surface. Because of the strong covalent C-C bonds on the SOC film surface and the reluctant nature of the chemical reaction between the polymer film surface and the CMP slurry chemicals, the mechanical-dominant CMP via rubbing between the CMP slurry abrasives and the SOC film surface was conducted, resulting in a relatively low polishing rate. For the first time, in our study, a catalyst for breaking the strong covalent C-C bonds on the SOC film surface during CMP was designed, by adding a ferric catalyst and an abrasive stabilizer to the CMP slurry. It is important to remember that the SOC film CMP should be conducted by both the mechanical-dominant CMP (i.e., rubbing between abrasives and the SOC film surface) and the chemical dominant CMP (i.e., a chemical reaction between the CMP slurry and the SOC film surface). First, among several ferric catalysts, iron(III) sulfate hydrate (Fe_2_(SO_4_)_3_H_2_O)) was selected, as it was demonstrated to be free of CMP slurry sedimentation and possessed the highest polishing rate at the inherent chemical pH of a ferric catalyst. Second, the dependency of the SOC film surface polishing rate on the ferric catalyst (iron(III) sulfate hydrate)) concentration was estimated to confirm the effect of the ferric catalyst on enhancing the SOC film polishing rate. Third, the dependency of the chemical bonding composition on the ferric catalyst concentration was observed by X-ray photoelectron spectroscopy (XPS) to delineate the presence of broken covalent C-C bonds on the SOC film surface during CMP. Fourth, the dependency of the CMP slurry adsorption degree (i.e., hydrophilicity) on the ferric catalyst concentration was investigated to find the chemical reaction degree between the CMP slurry and the SOC film surface during CMP, determining the absorption degree of the CMP slurry abrasives and the SOC film debris on the SOC film surface after CMP. Fifth, the dependency of the electrostatic force between the CMP slurry abrasive and the SOC film surface on the ferric catalyst concentration during CMP was tested to find the mechanical-dominant CMP property depending on the ferric catalyst concentration. Finally, based on both the chemical- and mechanical-dominant CMP characteristics of the SOC film surface depending on the ferric catalyst concentration in the CMP slurry, the mechanism by which the addition of ferric catalyst in the SOC film CMP slurry causes the strong covalent C-C bonds on the SOC film surface to break during CMP was proposed by the transition from carbon–carbon bonds to carbon–ferric iron(III)-carbon bonds on the SOC film surface during CMP.

## 2. Materials and Methods

### 2.1. Materials

A 300 nm-thick spin-on-carbon (SOC) film, composed of a polyarylene-ether block copolymer, was coated on a silicon substrate via spin-coating and annealing. The hardness of the SOC film was 0.63 Gpa, as shown in Appendix A. In this experiment, the 81.1 nm zirconia abrasives were synthesized with 40 nm zirconia dispersed in deionized (DI) water with polycarboxylic-acid type dispersant. The slurry was composed of zirconia abrasives using 0.05 wt%, 0~0.20 wt% of a ferric catalyst ([Fe_2_(SO_4_)_3_H_2_O]; Sigma Aldrich, St. Louis, MA, USA), 0.10 wt% of an abrasive stabilizer (i.e., picolinic acid; C_6_H_5_NO_2_; Sigma Aldrich, St. Louis, MA, USA), and DI water. Notably, the pH of the slurry containing only colloidal zirconia abrasive was 3.42, and the pH of the slurry composed of zirconia abrasives, the ferric catalyst, and the abrasive stabilizer decreased from 3.31 to 2.30 with the increase in the Fe_2_(SO_4_)_3_ concentration from 0.025 wt% to 0.20 wt%. Therefore, the pH of the slurry was titrated at 2.3 with HNO_3_ to exclude the effect of pH.

### 2.2. CMP Conditions

SOC film with a vertical structure of Si substrate/500 nm-thick SiO_2_ film/300 nm thick SOC film was cut into a 4 cm × 4 cm square. The CMP process of the SOC film surface was conducted using a CMP polisher (POLI-300, G&P Tech. Inc., Busan, Korea) implanted with a concentric-circle-grooved CMP pad (IC 1000, Dupont Co., Inc., Wilmington, DE, USA). Before polishing, the polishing pad was conditioned with a diamond disc and DIW for 30 min, and then two dummy wafers were polished prior to the main polishing of the SOC film surface. Pad conditioning was conducted in situ after each polishing for the various slurries. The applied head pressure was 6 psi, the rotation speed of the carrier holding the SOC film samples was 70 rpm, and the rotation speed of the table attached to the CMP pad was 70 rpm. The flow rate of the CMP slurry was fixed at 100 mL/min, and the polishing time was set to 30 s. After 30 s of CMP, all SOC film samples were buffed with DI water for 30 s to eliminate the remaining abrasives on the SOC film surface.

### 2.3. Characterization

The polishing rate of the SOC film was estimated by measuring the film thickness before and after the CMP using ellipsometry (V-VASE, J.A. Woollam Co., Inc., Lincoln, NE, USA). The secondary size and zeta potential of the zirconia abrasives in the CMP slurry and SOC film after CMP were analyzed using a particle analyzer (ELSZ2+, Otsuka Electronics Co., Inc., Osaka, Japan). The nano-scale (i.e., 81.1 nm in diameter) zirconia abrasives were observed using high-resolution transmission electron microscopy (HR-TEM, JEM-2010, JEOL Co., Inc., Tokyo, Japan) with an accelerating voltage of 200 kV. The surface roughness (average root mean square (RMS) roughness) of SOC film after polishing was estimated by atomic force microscopy (AFM, Park system, Suwon, Korea) with a 5 μm × 5 μm scan area. The contact angles were measured using a contact angle meter (GBX Instrument, DIGIDROP, Dublin, Ireland) by dropping 0.01 mL of DI water from the slurry on the SOC film surface after CMP. The chemical composition of the SOC film surface after CMP was characterized using XPS (X-ray photoelectron spectroscopy; K-Alpha+, Thermo Fisher Scientific Co., Inc., Waltham, MA, USA) at 12 keV and 6 mA with A1Kα (1486.6 eV). 

## 3. Results and Discussion

The CMP process is a dynamic cycling process involving a chemical reaction and a mechanical rubbing. To understand this dynamic cycling process, the mechanical properties (i.e., zeta-potential of ZrO_2_ abrasive, SOC film surface, and secondary ZrO_2_ abrasive size) determining the electrostatic force between the abrasives and the SOC film surface, the chemical properties (i.e., the chemical composition transformation of the SOC film surface), and the chemical–mechanical properties (i.e., the SOC film polishing rate) were characterized in detail.

### 3.1. Dependency of SOC Film Polishing Rate on Ferric Catalyst (Fe_2_(SO_4_)_3_H_2_O) Concentration

To enhance the polishing rate of the SOC film, a breakage of the C-C covalent bonds on the SOC film surface during CMP is introduced by adding a ferric catalyst (i.e., Fe_2_(SO_4_)_3_)) and an ZrO_2_ abrasive stabilizer (i.e., picolinic acid: C_6_H_5_NO_2_) to the CMP slurry. The SOC film CMP slurry was composed of 0.05 wt% ZrO_2_ abrasives of 81.1 nm diameter, and HNO_3_, Fe_2_(SO_4_)_3_H_2_O and C_6_H_5_NO_2_ at pH 2.3, as shown in the background TEM image of Figure 2. Among several ferric catalysts (i.e., (NH_4_)_5_[Fe(C_6_H_4_O_7_)_2_], (NH_4_)_3_[Fe(C_2_O_4_)_3_] H_2_O, Fe_2_(SO_4_)_3_H_2_O, C_11_H_18_N_2_O_8_4Fe, Fe(NO_3_)_3_9H_2_O, and K_4_Fe(CN)_6_), the highest polishing rate of the SOC film was achieved with Fe_2_(SO_4_)_3_H_2_O, as shown in Appendix A. The dependency of the SOC film-polishing rate on the ferric catalyst (i.e., Fe_2_(SO_4_)_3_H_2_O)) concentration was estimated, as shown in Figure 2. As a reference, the CMP slurry, without adding a ferric catalyst and an abrasive stabilizer, resulted in an SOC film-polishing rate of 27.7 nm/min. The polishing of the SOC film increased rapidly, from 27.7 to 202.3 nm/min, when the Fe_2_(SO_4_)_3_H_2_O concentration was enhanced from 0 to 0.05 wt%, as shown in Figure 2. Then, it decreased from 202.3 to 37.2 nm/min when the Fe_2_(SO_4_)_3_H_2_O concentration increased from 0.05 to 0.20 wt%. Hence, the polishing rate of the SOC film peaked at the Fe_2_(SO_4_)_3_H_2_O concentration of 0.05 wt%, i.e., Region 1 and 2. Otherwise, the secondary ZrO_2_ abrasive size was independent of the Fe_2_(SO_4_)_3_ concentration, i.e., ~150 nm. This result indicates that the addition of a 0.05 wt% ferric catalyst and abrasive stabilizer in the SOC film CMP slurry could enhance the polishing rate of the SOC film surface around seven times during CMP, presenting evidence of an enhancement of the chemical-dominant CMP characteristic in Region 1 via the C-C bond breakage of the SOC film surface, which will be discussed later. In addition, the polishing rate of the SOC film surface was further reduced with ferric catalyst concentration (i.e., >0.05 wt%), exhibiting a reduction in the mechanical-dominant CMP characteristic via enhancing the attractive electrostatic force between the ZrO_2_ abrasives and the SOC film surface in Region 2. 

### 3.2. Dependency of the C-C Bonds Breakage on the Ferric Catalyst (Fe_2_(SO_4_)_3_H_2_O) Concentration in the SOC Film CMP Slurry

The chemical bond composition of the SOC film surface immediately after CMP using slurries both with ferric catalyst (Fe_2_(SO_4_)_3_H_2_O) and without, as a reference, were analyzed by XPS as a function of the ferric catalyst concentration. This analysis was conducted to both find evidence of and quantify C-C bond breakage on the SOC film surface after CMP, depending on the ferric catalyst concentration in the CMP slurry. The C 1s spectra peaks of C–O, C–C and C-Fe (via C-C bond breakage) bonds were observed at 286.9, 285.4, and 284.0 eV, as shown in the insets of Figure 3a [20,21]. For the SOC film CMP slurry without a ferric catalyst, the relative C 1s spectra peak intensities for C–O, C–C and C-Fe bonds were 7214, 26,342, and 121 a.u., respectively. However, for the SOC film CMP slurry with a ferric catalyst, as the ferric catalyst concentration increased from 0 to 0.20 wt%, the relative C 1s spectra peak intensity for C–C decreased linearly from 26,342 to 22,014 a.u., while both C-O and C-Fe bonds increased linearly from 7214 to 9514 a.u. and from 121 to 5014 a.u., respectively, as shown in Figure 3b. In particular, at the same ferric catalyst concentration increase, the increase in the relative C 1s spectra peak intensity for C–Fe (i.e., 4893 a.u.) was higher than that for C-O (i.e., 2300 a.u.). The ferric catalyst (Fe_2_(SO_4_)_3_H_2_O) in the CMP slurry at a strong acidic pH (2.3) was well dissociated into Fe^3+^ and SO_4_^−2^. During CMP, Fe^3+^ chemically reacted with C-C bonds on the SOC film surface, thereby forming carbon metal complexes (i.e., C-Fe), while SO_4_^−2^, as an oxidant, oxidized C-C bonds, forming C-O bonds as a result. Note that SO_4_^−2^ is a well-known oxidant—the mechanism of formation of the carbon metal complex will be explained later. In addition, the O 2p spectra peaks of C–O and H-O–C bonds on the SOC film after CMP were observed at 531.2 and 532.6 eV, respectively, as shown in Figure 3c [22,23]. Both C-O-H and C-O bonds increased almost linearly from 158,213 to 208,157 a.u. and from 23,790 to 47,122 a.u., respectively, as shown in the insets of Figure 3c. In addition, at the same ferric catalyst concentration increase, the increase in the relative O 2p spectra peak intensity for C-O-H (i.e., 49,944 a.u.) was higher than that for C-O (i.e., 23,332 a.u.). Again, these results indicate that SO_4_^−2^ oxidized C-H as well as C-C bonds, resulting in the formation of C-O-H and C-O bonds on the SOC film surface after CMP. It is important to remember that the SOC film surface was composed of a polyarylene-ether block copolymer including C-C, C=C, O-C, and C-H bonds. Moreover, the Fe 2p spectra peaks of Fe2p_1/2_(FeO), Fe(II)_3/2_, and Fe2p_3/2_(FeO) bonds on the SOC film after CMP were found at 723.1, 715.2, and 710.2 eV, respectively, as shown in Figure 3d [24]. The Fe 2p spectra peak intensities of Fe2p_1/2_(FeO), Fe(II)_3/2_, and Fe2p_3/2_(FeO) bonds almost linearly increased from 115 to 3477 a.u., 104 to 3204 a.u., and 97 to 2904 a.u., respectively, as shown in Figure 3e. These results demonstrate that during CMP, Fe^3+^ chemically reacted with O-C bonds on the SOC film surface, generating FeO bonds, and diffused into C-C, C=C, O-C, and C-H bonds without chemical reaction through a rubbing process between abrasives and the SOC film surface, producing interstitial Fe^3+^. Finally, the S 2p spectra peaks of the S2p_3/2_ bond on the SOC film after CMP were shown at 161.8 eV, as shown in Figure 3f [25]. The S 2p spectra peak intensities of the S2p_3/2_ bond almost linearly increased from 115 to 3477 a.u., as shown in the inset of Figure 3f. This result means that S^2−^ ions diffused into C-C, C=C, O-C, and C-H bonds without chemical reaction through a rubbing process between abrasives and the SOC film surface, producing interstitial S^2−^ ions. Therefore, during the SOC film surface CMP, using the CMP slurry, including nanoscale (i.e., 45 nm in diameter) ZrO_2_ abrasives, a ferric catalyst (i.e., Fe_2_(SO_4_)_3_)) and an abrasive stabilizer (i.e., picolinic acid: C_6_H_5_NO_2_), the chemical compositions of the SOC film surface were transformed from C-C, C=C, O-C, and C-H bonds into C-O, C-Fe, C-O-H, FeO bonds, Fe^3+^ ions, and S^2−^ ions, increasing the polishing rate of the SOC film surface with the ferric catalyst (i.e., Fe_2_(SO_4_)_3_)), as shown in Figure 2. In particular, comparing the spectra peak intensities for C-O, C-Fe, C-O-H, and FeO bonds, Fe^3+^ ions, and S^2−^ ions, although the higher sequence of the spectra peak intensities on the SOC film surface after CMP was followed by C-O-H, C-O, C-Fe, FeO, interstitial Fe^3+^, and S^2−^ ions; the formation of C-Fe bonds via C-C bond breakage enhanced the polishing rate of the SOC film surface, as it resulted in C-C bond breakage rather than the oxidation of C-C bonds, such as C-O-H and C-O bonds. Thereby, the polishing rate of the SOC film surface would increase with the ferric catalyst (i.e., Fe_2_(SO_4_)_3_) concentration in the SOC film CMP slurry, mainly influencing the chemical properties of the SOC film CMP. Furthermore, the formation degree of the carbon–iron complex on the SOC film surface during CMP would be limited by the decomposition degree of the ferric catalyst into ferric ions (Fe^3+^) that depends on the ferric catalyst type and pH in the CMP slurry. At an acidic pH (i.e., 2.3), Fe_2_(SO_4_)_3_H_2_O C_11_H_18_N_2_O_8_4Fe, and Fe(NO_3_)_3_9H_2_O could achieve the maximum decomposition degree of the ferric catalyst into ferric ions (Fe^3+^). Among them, Fe_2_(SO_4_)_3_H_2_O demonstrated the highest polishing rate of the SOC film, indicated that the higher sequence of the maximum decomposition degree would be followed by Fe_2_(SO_4_)_3_H_2_O, C_11_H_18_N_2_O_8_4Fe, and Fe(NO_3_)_3_9H_2_O, as shown in Appendix A.

### 3.3. Dependency of the Mechanical Properties (i.e., Electrostatic Force between Abrasive and SOC Film Surface and Absorption Degree of CMP Slurry) on the Ferric Catalyst (i.e., Fe_2_(SO_4_)_3_) Concentration

As the polishing rate of the SOC film surface peaked at a specific ferric catalyst concentration, as shown in Figure 2, the dependency of the chemical property (i.e., C-C bonds breakage) on the ferric catalyst (Fe_2_(SO_4_)_3_H_2_O) concentration could not completely explain the dependency of the SOC film surface polishing rate on the ferric catalyst concentration. Hence, the dependency of the mechanical properties (i.e., electrostatic force between abrasive and the SOC film surface and absorption degree of CMP slurry) on the ferric catalyst concentration was estimated for the SOC film surface CMP, as shown in Figure 4. When the ferric catalyst concentration increased from 0 to 0.04 wt%, the zeta-potential of the SOC film surface significantly decreased from −14.13 to 0 mV. Then, it increased considerably, from 0 to +26.91 mV, when the ferric catalyst concentration increased from 0.04 to 0.20 wt%. This result demonstrated that there are two regions of the ferric catalyst concentration, i.e., Region 1 (negatively charged SOC film surface) and Region 2 (positively charged SOC film surface). Otherwise, the zeta-potential of the ZrO_2_ abrasives in the CMP slurry decreased slightly, from −3.42 to −1.21 mV, when the ferric catalyst concentration increased from 0 to 0.20 wt%. As a result, for Region 1, with a ferric catalyst concentration of 0~0.04 wt%, the repulsive force between the negatively charged ZrO_2_ abrasives and the negatively charged SOC film surface decreased abruptly from 48.32 to 0 abs. This result predicts that the polishing rate of the SOC film increases rapidly with the ferric catalyst concentration, as shown in Region 1 of Figure 2. It is important to remember that, generally, the polishing rate of the SOC film surface increases with a decreasing repulsive force between the ZrO_2_ abrasives and the SOC film surface. [26,27,28,29]. Otherwise, for Region 2, with a ferric catalyst concentration of 0.04~0.20 wt%, the attractive force between the negatively charged ZrO_2_ abrasives and positively charged SOC film surface increased notably, from 0 to 32.67 abs. This result demonstrates that the polishing rate of the SOC film increases considerably with ferric catalyst concentration, and would show the inverse dependency of the SOC film surface polishing rate on the ferric catalyst concentration in Region 2 of Figure 2. In general, the polishing rate of the SOC film surface increases with the attractive force between the ZrO_2_ abrasives and the SOC film surface. [26,27,28,29]. Therefore, for Region 2, the increase in the SOC film surface polishing rate with the ferric catalyst concentration could not be understood from the increase in both the chemical property (i.e., C-C bonds breakage) and the mechanical property (i.e., electrostatic force between the ZrO_2_ abrasive and the SOC film surface).

To precisely understand the dependency of the SOC film surface polishing rate on the ferric catalyst concentration for Region 2, the absorption degree of the SOC film CMP slurry on the polished SOC film surface was measured as a function of the ferric catalyst concentration, as shown in Figure 5. As a reference, the DIW of 0.01 mL was dropped on the unpolished SOC film surface (i.e., negatively charged surface), and the contact angle of DIW was 77.45°. This result implies that the unpolished SOC film has a strong hydrophobic surface. However, when the SOC film surface CMP slurry of 0.01 mL was left on the polished SOC film surface, the contact angle of the SOC film surface CMP slurry decreased notably, from 62.71 to 42.18, when the ferric catalyst concentration was 0~0.20 wt%. This result indicates that the degree of absorption (i.e., hydrophilicity) of the SOC film CMP slurry on the polished SOC film surface enhances the ferric catalyst concentration. Considering Stoke’s law for abrasive moving behavior in a fluid (i.e., the SOC film surface CMP slurry), as shown in Appendix A, during CMP, a higher absorption degree of the SOC film CMP slurry on the polished SOC film surface would lead to less movement in and out of the ZrO_2_ abrasives on the CMP pad [30]. Thus, the polishing rate of the SOC film surface decreased with increasing ferric catalyst concentration, as the absorption degree of the SOC film CMP slurry on the polished SOC film surface increased significantly with the ferric catalyst concentration, demonstrating why the polishing rate of the SOC film surface decreased with increasing ferric catalyst concentration for Region 2, as shown in Figure 2. In summary, the polishing rate of the SOC film surface enhanced rapidly with the ferric catalyst concentration for Region 1, as shown in Figure 2, mainly related to the chemical properties (i.e., C-C bond breakage). Otherwise, it decreased with increasing ferric catalyst concentration for Region 2, as shown in Figure 2, predominantly associated with the mechanical property (i.e., the absorption degree of the CMP slurry rather than the electrostatic force between the ZrO_2_ abrasive and the SOC film surface). Furthermore, to confirm the cleaning performance, the absorption degree of the CMP slurry after dipping and loading-up the SOC film wafer in the CMP slurry was observed as a function of the ferric concentration CMP slurry, as shown in Appendix A. Note that, in general, the post cleaning after CMP had been conducted by a DI and diluted HF spray cleaning on the vertically loaded wafer so that a free of the remaining CMP slurry on the SOC film surface has been required after dipping and loading-up the SOC film wafer in the CMP slurry. Fortunately, none of the remaining CMP slurry on the SOC film surface were found after dipping and loading-up the SOC film wafer in the CMP slurry, as shown in Appendix A. This result indicates that the remaining CMP slurry on SOC film surface after CMP would be completely cleaned, although the formation of carbon–iron complex transferred the surface zeta potential of the SOC film from the negative-charged surface to the positive-charged surface.

### 3.4. C-C Bond Breakage Mechanism of the SOC Film Surface during CMP Using Ferric Catalyst (i.e., Fe_2_(SO_4_)_3_) Concentration

During the SOC film surface CMP, four different types of surface chemical reaction could happen, based on the chemical compositions of the polished SOC film surface, depending on the ferric catalyst concentration, as shown in Figure 3, i.e., C-O-H, C-O, C-Fe (i.e., C=Fe=C), Fe=O, interstitial Fe, and interstitial S. First, the C–H bonds on the SOC film surface (i.e., polyarylene-ether block copolymer) were dissociated by yielding carbon radicals and an intermediate iron complex containing a hydroxyl group (OH) on the SOC film surface, as shown in **a** and **b** of Figure 6a [31]. Afterwards, the hydroxyl group of the intermediate iron complex was transferred to the carbon radicals on the SOC film surface and the remaining Fe atom was oxidized back to its original form (i.e., Fe=O). This reaction was repeated, eventually producing C-O-H bonds on the SOC film surface with 532.6 eV in binding energy, as shown in **b** and **c** of Figure 6a. Second, the C-H bonds on the SOC film surface were dehydrated, producing H_2_ and two carbon radicals, as shown in **a** and **b** of Figure 6b. Afterwards, two dehydrated =C- bonds (i.e., carbon radicals) were attached to O atoms, producing C-O bonds on the SOC film surface with 286.9 and 531.2 eV in binding energy, as shown in **b** and **c** of Figure 6b. Third, Fe atoms were inserted into C-H bonds on the SOC film surface via an oxidative addition reaction, as shown in **a** and **b** of Figure 6c [31,32,33,34,35]. The second C–H bond additions were located at the adjacent C–H bonds, known as β-hydrogen elimination, as shown in **b** and **c** of Figure 6c [36]. Then, two hydrogen atoms bonded with Fe atom were separated by generating a molecule of hydrogen. Finally, the C–C bonds bridged by the Fe complexes were broken, yielding C-C bond breakage (i.e., C=Fe=C) on the SOC film surface with 284 eV in binding energy, as shown in **c** and **d** of Figure 6c [37]. Fourth, Fe=O, Fe^3+^ ions, and S^2−^ ions were inserted into the SOC film surface via rubbing between the ZrO_2_ abrasives and the SOC film surface, with 710, 715, and 162 eV in binding energy, respectively, generating interstitial Fe=O, Fe^3+^ ions, and S^2−^ ions on the SOC film surface, as shown in Figure 6d. Among the transformed surface bonding states (i.e., C-O-H, C-O, and C=Fe=C) on the SOC film surface during CMP using the ferric catalyst (i.e., Fe_2_(SO_4_)_3_), the C=Fe=C bonds presented the lowest binding energy, and thus were easily broken during rubbing between the ZrO_2_ abrasives and the SOC film surface, enhancing the polishing rate of the SOC film surface. However, the transformation of the SOC film surface from C-C, C=C, O-C, and C-H bonds to C-O-H, C-O, C=Fe=C, Fe=O, interstitial Fe^3+^ ions, and interstitial S^2−^ ions on the SOC film surface during CMP significantly enhanced the absorption degree (i.e., hydrophilicity) of the SOC film surface CMP slurry, reducing the polishing rate of the SOC film surface. As a result of the C-C bond breakage, as well as the absorption degree of the CMP slurry, the polishing rate of the SOC film surface (i.e., 202.3 nm/min) peaked at a ferric catalyst concentration of 0.05 wt%, as shown in Figure 2. 

## 4. Conclusions

As the SOC film (i.e., polyarylene-ether block copolymer) surface, as a hard-mask material, was principally composed of strong carbon covalent bonds (i.e., C-C, C=C, O-C, and C-H bonds), a breakage of the strong carbon covalent bonds is necessary for the planarization of a huge topography of the SOC film surface during CMP. The mechanical-dominant CMP (i.e., mechanical rubbing between the ZrO_2_ abrasives and the SOC film surface) itself could not perform a perfect removal of such high topography. Thus, an additional chemical-dominant CMP (i.e., transformation of a hard carbon covalent bond into a soft carbon–metal complex on the SOC film surface) was introduced by adding a ferric catalyst (i.e., Fe_2_(SO_4_)_3_) in the SOC film CMP slurry. During CMP, the SOC film surface was transformed from a hard surface, with C-C, C=C, O-C, and C-H bonds, to a soft surface, with C-O-H, C-O, C=Fe=C, Fe=O, interstitial Fe^3+^ ions, and interstitial S^2−^ ions. In particular, the C-C bond breakage (i.e., C=Fe=C bonds) on the SOC film surface enhanced the polishing rate of the SOC film surface; as a result, the polishing rate increased with the ferric catalyst (i.e., Fe_2_(SO_4_)_3_) concentration. These surface transformations enhanced the chemical-dominant CMP property during the SOC film surface CMP. However, during CMP, the SOC film surface transformation from C-C, C=C, O-C, and C-H bonds to C-O-H, C-O, C=Fe=C, Fe=O, interstitial Fe^3+^ ions, and interstitial S^2−^ ions transformed the SOC film surface from a strongly hydrophobic to a hydrophilic surface, i.e., the absorption degree (i.e., hydrophilicity) of the SOC film CMP slurry on the polished SOC film surface increased noticeably with the ferric catalyst concentration. Thus, the polishing rate of the SOC film surface was notably decreased with increasing ferric catalyst concentration. The absorption degree of the SOC film surface CMP slurry on the polished SOC film surface decreased greatly, owing to the mechanical-dominant CMP property. Because of a trade-off in the polishing rate between the chemical-dominant and mechanical-dominant CMP properties, depending on the ferric catalyst concentration, the polishing rate of the SOC film surface peaked at a specific ferric catalyst concentration (i.e., 0.05 wt%). This study can be expanded for another organic polymer-film surface CMP, i.e., amorphous carbon film as a hard-mask of semiconductor lithography and polyimide as an insulator material in an OLED display. For these new CMP applications, further studies on the design of a CMP slurry are required. In addition, for a real industrial application, such as logic devices and DRAM, when our proposed CMP slurry was used for the SOC film CMP, the dependency of erosion on the pattern density would be essentially investigated.

## Figures and Tables

**Figure 1 nanomaterials-12-00969-f001:**
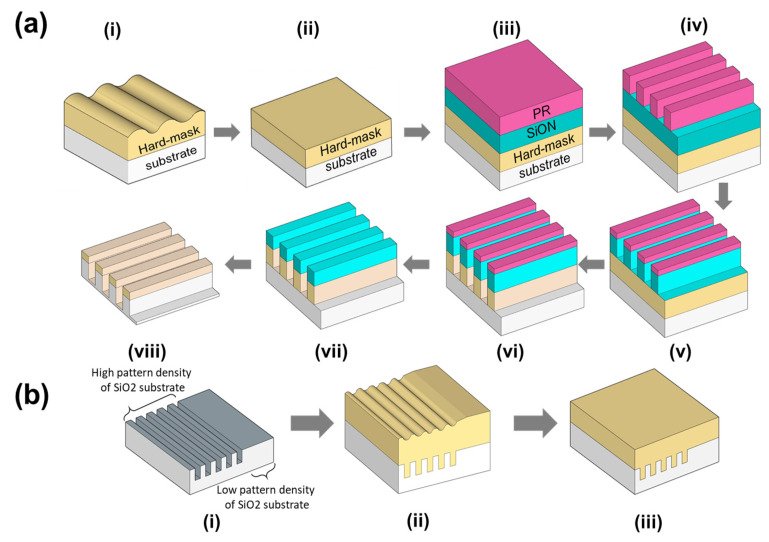
(**a**) Fabrication process flow of the lithography and etching using the SOC film hard-mask followed by CMP. (**b**) SOC film surface CMP.

**Figure 2 nanomaterials-12-00969-f002:**
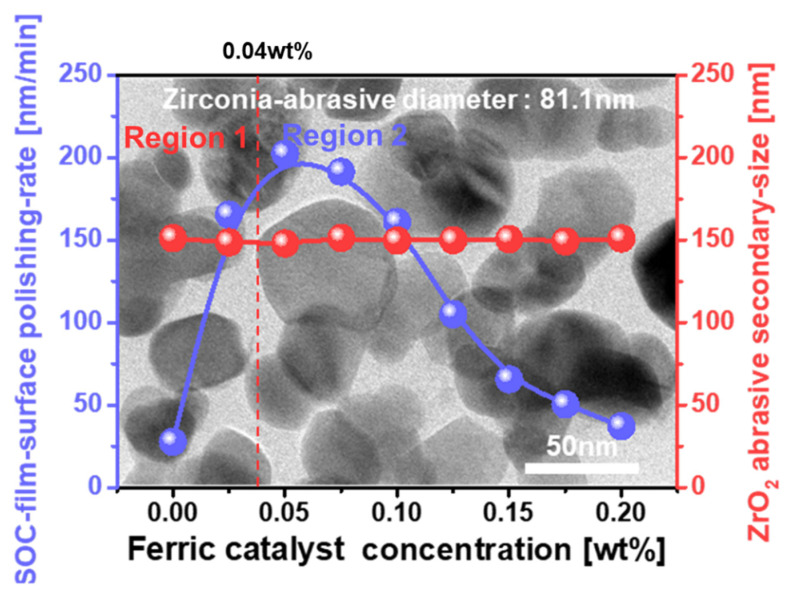
Effect of the ferric catalyst (Fe_2_(SO_4_)_3_H_2_O)) on the SOC film polishing rate and the ZrO_2_ abrasive secondary size, where a background SEM image presents the ZrO_2_ abrasive morphology, with a crystalline round shape and a diameter of ~81.1 nm.

**Figure 3 nanomaterials-12-00969-f003:**
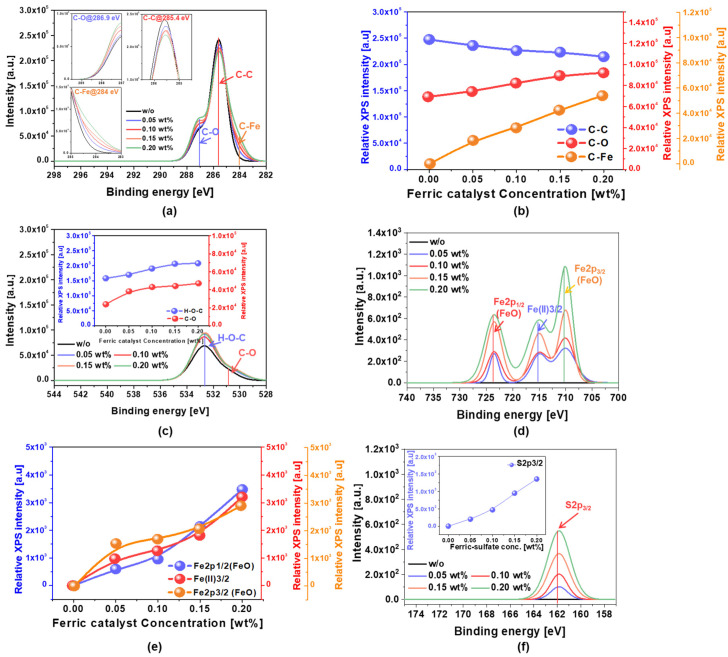
Dependency of the chemical compositions on the ferric catalyst concentration for the polished SOC film surface after CMP, analyzed by XPS. (**a**) C 1s spectra, (**b**) relative XPS intensity for C 1s spectra depending on the ferric catalyst concentration, (**c**) O 1s spectra, (**d**) Fe 2p spectra, (**e**) relative XPS intensity for Fe 2p spectra depending on the ferric catalyst concentration, and (**f**) S 2p spectra.

**Figure 4 nanomaterials-12-00969-f004:**
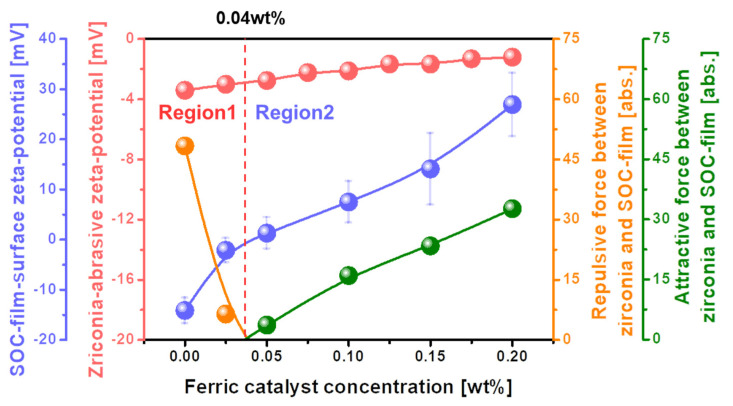
Relative electrostatic force between the ZrO_2_ abrasives and the polished SOC film surface, depending on the ferric catalyst (Fe_2_(SO_4_)_3_H_2_O)) concentration in the SOC film CMP slurry.

**Figure 5 nanomaterials-12-00969-f005:**
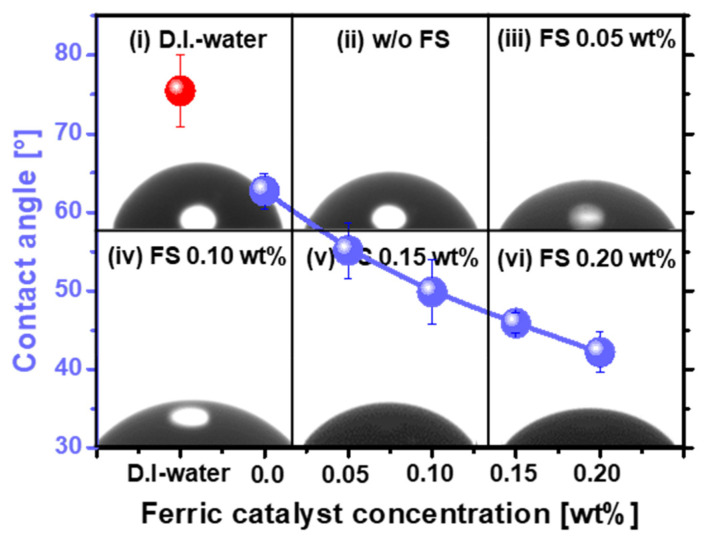
Adsorption degree (i.e., contact angle) of the SOC film surface CMP slurry on the polished SOC film surface, depending on the ferric catalyst (Fe_2_(SO_4_)_3_H_2_O)) concentration.

**Figure 6 nanomaterials-12-00969-f006:**
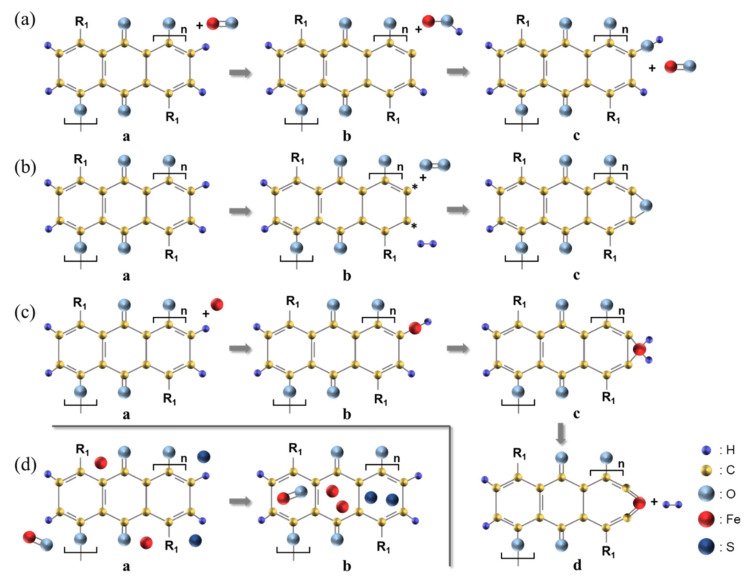
Transformation of the surface bonding states on the SOC film surface during CMP using the ferric catalyst (i.e., Fe_2_(SO_4_)_3_). The formation of (**a**) C-O-H, (**b**) C-O, (**c**) and C=Fe=C (C-C bond breakage), and (**d**) Fe=O, interstitial Fe, and interstitial S bond on the SOC film surface.

## Data Availability

Data can be available upon request from the authors.

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
