# Peer review of "Surface Transformation of Spin-on-Carbon Film via Forming Carbon Iron Complex for Remarkably Enhanced Polishing Rate"

_nanomaterials, 2022, doi:10.3390/nano12060969_

Round 1
Reviewer 1 Report
This contribution tackle of the removal mechanisms of spin-on carbon thin films during chemical mechanical planarization. Extensive surface characterization works were performed to elucidate the observed dependence of removal rates upon ferric catalyst concentration, based on combined electrostatic arguments, bond-breaking mechanisms, and surface adsorption.
The is a fundamental study on a technologically important material for semiconductor processing that would provide insights into surface removal mechanisms which could also provide potential guidance to CMP slurry formulation. A welcoming study would entice more to come in this field.
Author Response
Thank you for your comment.

Reviewer 2 Report
This manuscript is an interesting work dealing with surface sransformation of Spin-On-Carbon film via forming carbon iron complex for enhanced polishing rate.
THE WHOLE WORK IS INTERESTING.
POINTS FOR IMPROVEMENT:
- Does the formation of carbon-iron complex has any undesired effects?
- What are the limitations of the method? A relative discussion is required.
- What are the main problems for industrial application?
- A Patent literature review for similar work could be made.
In my opinion this work could be published after revision.
Author Response
Thank you for your comment.
